# Evans syndrome in children below 13 years of age – A nationwide population-based cohort study

**Nikolaj Mannering**[1,2]*, **Dennis Lund Hansen**[1,2], **Henrik Frederiksen**[1,2]

**1** Department of Hematology, Odense University Hospital, Odense, Denmark, **2** Department of Clinical Research, University of Southern Denmark, Odense, Denmark

* nikolaj.mannering2@rsyd.dk

## Abstract

Evans syndrome is defined by autoimmune haemolytic anaemia and immune thrombocytopenia occurring in the same patient. Although known to be rare the frequency and prognosis of Evans syndrome in children is unknown, and only few registry-based studies are available. The epidemiology and prognosis of Evans syndrome in patients above 13 years of age has recently been investigated. In this age group both incidence and prevalence of Evans syndrome increased during the study period and median survival was just 7.2 years. Using Danish health registries and the same approach, we identified 21 children below 13 years of age with Evans syndrome during 1981–2015. Patients with Evans syndrome were age–and sex matched with children both from the general population, and with patients with either autoimmune haemolytic anaemia or immune thrombocytopenia. The incidence of Evans syndrome ranged between 0.5 and 1.2/1,000,000 person-years. Prevalence was 6.7 and 19.3/1,000,000 in 1990 and 2015 respectively. Hazard ratio for death was 22 fold higher for children with ES compared to matched children from general population, and was also elevated compared to children with autoimmune haemolytic anaemia or immune thrombocytopenia. We conclude that pediatric ES is very rare and associated with elevated mortality. However, despite the nationwide study and a long and complete follow-up, results are imprecise due to the rarity of this disorder.

## Introduction

Evans syndrome (ES) defines a concurrent or sequential occurrence of autoimmune haemolytic anaemia (AIHA) and immune thrombocytopenia (ITP), caused by immune reactions against erythrocytes and platelets [1].

Treatment options in ES are immunosuppressive drugs such as corticosteroids and rituximab, but also splenectomy remains a treatment option. In contrast to ITP in childhood, ES often follow a chronic or relapsing-remitting course. ES is associated with higher need of treatment and higher mortality compared to ITP and AIHA [2–5].

Causes of death in childhood ES are reported to be both treatment-related (e.g. increased risk of infections), and disease-related factors (e.g. major haemorrhage) [6, 7].

**Data Availability Statement:** Data cannot be shared publicly because of Danish law. Data are available from the Statistics Denmark Institutional Data Access / Ethics Committee (contact via

https://www.dst.dk/en#) for researchers who meet the criteria for access to confidential data.

**Funding:** This project was part of a PhD fellowship (DLH), which was supported by the University of Southern Denmark (SDUSF-2015-202-(459)) https://www.sdu.dk/en/, The Region of Southern Denmark (16/13496) https://www.regionsyddanmark.dk/wm228983, and study grants from Alexion Pharma Nordic AB https://alexion.com/SelectCountry/Denmark, The A.P. Møller and Chastine Mc-Kinney Møller Foundation (17-L-0334) https://www.apmollerfonde.dk/ansoegning/laegefonden/, and Novartis Healthcare https://www.novartis.com/. The funders had no role in study design, data collection and analysis, decision to publish, or preparation of the manuscript.

**Competing interests:** The project was funded by Alexion Pharma Nordic AB and Novartis Healthcare. None of the authors have any financial interests of any kind in any of the companies, employment in e.g. Advisory Boards or consultancy work. None of the authors are currently participating in any kind of drug development related to any of the companies. This does not alter our adherence to PLOS ONE policies on sharing data and materials.

However, few studies of ES in children are available, most of them with smaller groups of patients due to the rarity of the disorder.

No studies have to our knowledge estimated prevalence of childhood ES and incidence has only been estimated by one previous study [6].

Here we assess both frequency and prognosis in Danish children below 13 years of age with ES, and compare their prognosis both with children from the general population and with children who have either isolated AIHA or ITP.

## Methods

Patients with ES were identified in Danish Health Registries. Denmark provides tax funded universal health care for all inhabitants, and all hospitals report to the National Danish Health Registries and other administrative registries [8]. We linked data from the Danish Civil Registration System [9], Danish National Patient Registry (DNPR) [10] and the Danish Registry of Causes of Death [11]. The methods applied were described in details in a recent study by us [7].

In short, ES was defined as the cumulative registration of ITP and AIHA in the same patient. The registration of the diagnoses could be sequential (85.7% of the cases) or simultaneous (14.3% of the cases). All patients with ES below 13 years of age diagnosed between 1981–2015 were included in analyses. Patients were included on the first date at which both ITP and AIHA had been recorded.

All patients with ES were age- and sex-matched with up to 50 individuals from the general population, and up to 4 patients with only AIHA or ITP. Comparisons were allotted the same day of diagnosis as their index patient.

Secondary ES was defined as the registration of associated diagnoses (such as cancer and rheumatic disorders) in the DNPR before or up to one year following ES diagnosis [7].

### Analysis

Patients and their matched individuals were followed from inclusion until 20 years post-diagnosis, emigration, death, or the end of December 2017, whichever came first. Cause of death was categorized as cardiovascular, bleeding, hematological cancer, hemolysis, infection, solid cancer and other causes. The Kaplan-Meier estimator was used to calculate survival. Unadjusted Cox proportional hazard regression was used to calculate hazard ratios (HR) for death between the ES and comparisons, and for AIHA and ITP compared to general population. A p-value <0.05 was considered statistically significant.

The incidence rate per 1,000,000 person-years was calculated for the periods: 1981–1990 and 2006–2015. We used cumulative incidences of ES during the periods, divided by the cumulative population below 13 years of age at risk during the same periods. The one-year-period-prevalence of ES was calculated for the years: 1990 and 2015, using the population less than 13 years of age as denominator. Analyses were performed using Stata 15.1 (StataCorp, TX, USA).

### Ethics statement

All data used in this retrospective study were completely anonymized, and no detailed patient information e.g. social security numbers or detailed journal information, were available. Permission from The Danish Ethics Committee was therefore not necessary, neither were any written consents from any of the participating patients.

## Results

We identified 159 and 3160 patients below 13 years of age with AIHA and ITP respectively, in the period 1981–2015. Of these, 21 patients fulfilled study criteria for ES. Mean age at ES diagnosis was 4.7 years and seven (33%) were female. Overall observation time was 16,287 person-years, mean observation time was 13.3 person-years. We also identified 68 age–and sex-matched children with only AIHA (mean age: 3.8 years) and similarly 86 children with only ITP (mean age: 4.8 years) (Table 1).

Secondary cases of ES were diagnosed in four (19%) patients, all secondary to haematological malignancy. In comparison, defined secondary causes were diagnosed in six (8.8%) patients with AIHA and six (7.0%) patients with ITP.

The median lead time between AIHA / ITP diagnosis was 0.3 years. The proportion of ES among patients with AIHA was 11.7% (21/159) (95% confidence interval (95% CI)): 7.4–17.3) and 0.7% (21/3160) (95% CI: 0.4–1.0) among patients with ITP.

The incidence of ES in our study period was 0.5 (95% CI: 0.1–1.3) and 1.2 (95% CI: 0.6–2.2) per 1,000,000 person-years during 1981–1990, and 2006–2015, respectively. Prevalence was 6.7 (95% CI: 2.2–15.7) and 19.3 (95% CI: 11.1–31.4) per 1,000,000 persons in 1990 and 2015, respectively.

Splenectomy was performed in four (19%) patients. Splenectomy rates were higher in patients with ES, both compared to the general population, and to patients with AIHA or ITP (Table 1). Of the four patients receiving splenectomy, three had the procedure done before the year 2000. Furthermore, all patients had the procedure done no more than 5 years after the Evans diagnosis.

During the study, two patients with ES died, both within the first year following diagnosis. One of the patients suffered from secondary ES. The causes of death in patients with ES were registered as bleeding or malignancy.

**Table 1. Basic characteristics of patients below 13 years of age with Evans syndrome and comparison groups.** (please note that all intervals in parenthesis following numbers in the table are 95% confidence intervals).

| | Evans Syndrome (n = 21) (95% CI) | General population comparison population (n = 1,049) (95% CI) | Matched AIHA population (n = 68) (95% CI) | Matched ITP population (n = 86) (95% CI) |
|---|---|---|---|---|
| Females (%) | 33.3 (14.6–57.0) | 33.3 (30.4–36.2) | 38.2 (26.7–50.8) | 32.6 (22.8–43.5) |
| Death (%) | 9.5 (1.2–30.4) | 1.0 (0.5–1.7) | 5.9 (1.6–14.4) | <3.5 (0.0–6.3) |
| Age at diagnosis (mean) | 4.7 (3.3–6.0) | 4.7 (4.5–4.8) | 3.8 (2.9–4.6) | 4.8 (4.1–5.4) |
| Splenectomy (%) | 19.0 (5.4–41.9) | 0.1 (0.0–0.5) | 4.4 (0.9–12.4) | 1.2 (0.0–6.3) |
| ES proportion of ITP (%) | 0.7 (0.4–1.0) | | | |
| ES proportion of AIHA (%) | 11.7 (7.4–17.3) | | | |
| Secondary (%) | 19.0 (5.4–41.9) | | 8.8 (3.3–18.2) | 7.0 (2.6–14.6) |
| ITP before AIHA (%) | 33.3 (14.6–57.0) | | | |
| Simultaneous (%) | 14.3 (3.0–36.3) | | | |
| AIHA before ITP (%) | 52.4 (29.8–74.3) | | | |
| Mean time between AIHA/ITP diagnoses (years) | 1.6 (0.6–2.7) | | | |
| Median and IQR between diagnoses (years) | 0.3 (0.1–1.7) | | | |
| 1-year survival (%) | 90.5 (67.0–97.5) | 100.0 (n/a) | 97.0 (88.7–99.3) | 98.8 (92.0–99.8) |

Basic characteristics of patients below 13 years with Evans syndrome (ES). The comparison population is random age- and sex-matched comparisons from the general population. Matched AIHA population: age- and sex-matched comparisons with only autoimmune haemolytic anaemia (AIHA). Matched ITP population: age- and sex-matched comparisons with only immune thrombocytopenia (ITP). n/a: not applicable.

In the AIHA comparison group four patients died, and in the ITP comparison group one patient died. Causes of death were registered as haemolysis or were not recorded.

Fig 1 depicts that mortality was higher in patients with ES, compared to patients with AIHA or ITP. The HR for ES mortality compared with general population was 22.3 (95% CI: 4.3–115.1; p <0.001) fold increased, while HR for death in patients with AIHA and patients with ITP were 11.8 (95% CI: 3.2–44.0; p <0.001) and 2.5 (95% CI: 0.3–21.0; p = 0.412) compared to the general population respectively.

## Discussion

Diagnosing ES relies on diagnostic criteria for both ITP and AIHA. In this process, other factors such as infections (e.g. Epstein-Barr virus (EBV), human immunodeficiency virus (HIV)), drug-induced cytopenias, immune defects and associated autoimmunity (e.g. systemic lupus erythematosus (SLE)), must be pursued [6]. Our study is based on routinely collected registry data. However, the validity of the ITP and AIHA diagnoses is high in Danish registries [7, 12].

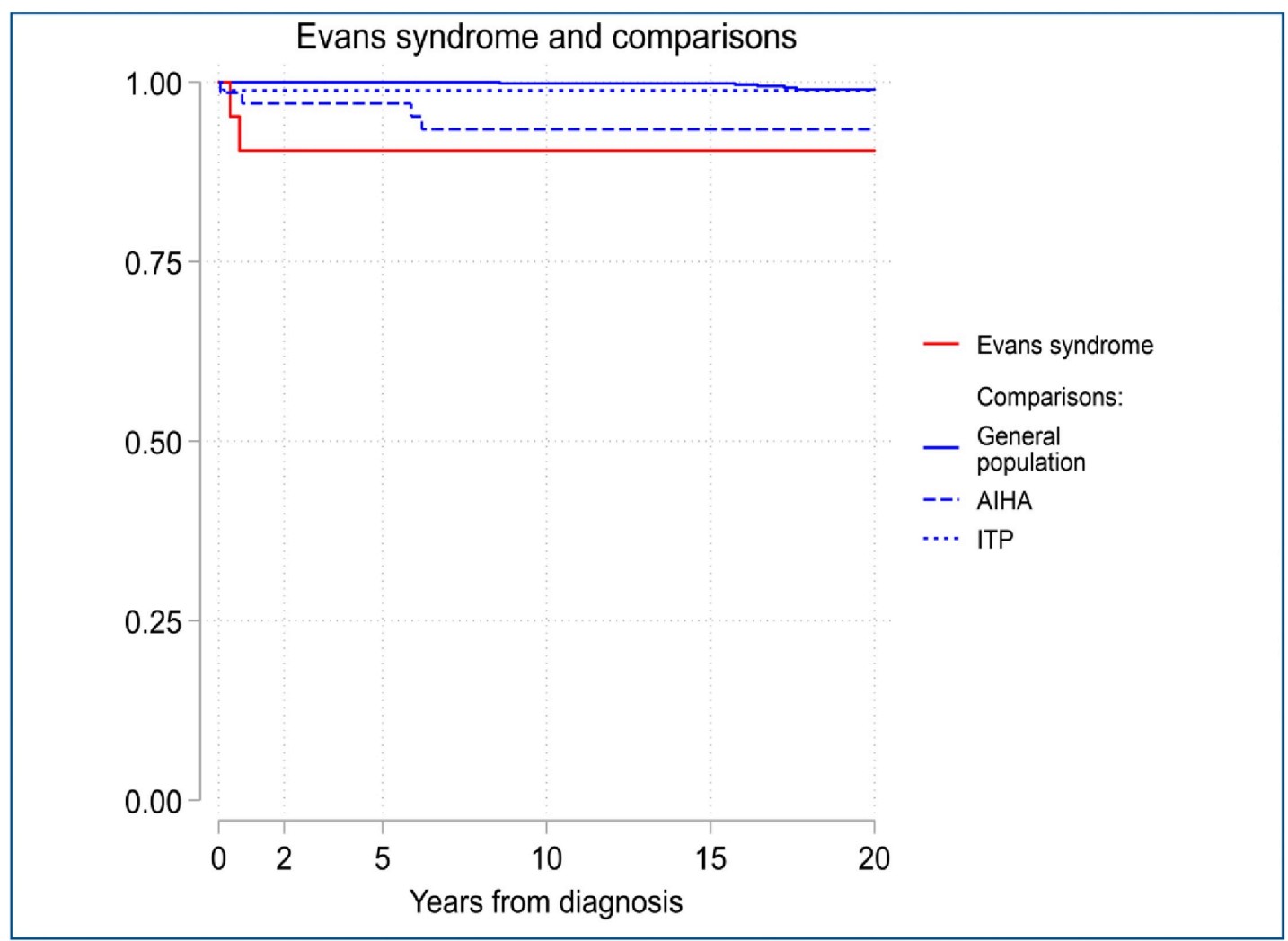

**Fig 1. Kaplan-Meier survival curves for patients below 13 years with Evans syndrome and comparisons.**

We found that all four cases of secondary ES were related to haematological malignancy. Our results differ from another study where none of the secondary ES were related to malignancy [6].

We found that ES was more predominant in boys compared to girls with a sex ratio of 2:1. This is in contrast to the general distribution of autoimmune diseases in adulthood, where females are more predominant [13], and also to ES in adults where the overall sex ratio is 1:1 [7]. The mechanism behind this phenomenon is unknown. A possible explanation to this could be differences in the pathophysiogical mechanisms preceding ES in children and adults respectively. Viral infections often precedes ES in young children [14], while adult patients may suffer from underlying haematological cancer (e.g. CLL [15]).

The reasons for the male predominance in very young patients with ES remain elusive. Genetically conditioned sex-differences, resulting in a more sensitive and responsive immune system, with a more profound release of cytokines (e.g. interferon-gamma [4]) following a viral infection may contribute to this [16].

Of note, ITP among boys below 5 years of age has a similar sex ratio of 2:1 as in ES [17].

Our data are collected during 35 years, and is one of the first to report an estimate of both incidence and prevalence of ES in children. Despite of this, we managed to identify only 21 patients emphasizing that pediatric ES is a very rare entity. This makes conclusions based on clinical experience weak, and studies will often lack power. Larger cohorts than ours have reached comparable conclusions with an annual incidence of 10 in French children less than 18 years of age in a French study, corresponding an annual incidence of approximately 0.7/ 1,000,000 [6]. The French study reports an overall mortality rate of 10% and similar sex distribution in line with our results.

## Survival

We found that ES in children below 13 years of age is associated with reduced long term survival and increased rate of splenectomy. Most splenectomies were performed before the year 2000. This is probably due to limited medical treatments options in this period (e.g. rituximab), and corresponds with splenectomy being a more frequent treatment early in our study period [18]. However, our data are based on few numbers of patients. Splenectomy is usually reserved as a 3rd or later line therapy in pediatric AIHA, ITP or ES, and with achieved complete remission rates greatly depending on disease with the poorest responses in ES [3, 18].

Although one out of two patients who died also suffered from a haematological cancer, one of the fatalities had primary ES indicating that even without an underlying disorder, pediatric ES is a severe disease.

Our survival analyses indicate that particular in AIHA, mortality and the occurrence of ES is increased, most pronounced in secondary ES cases. These results are in line with our previous study of adults with ES where particularly patients with secondary ES had a dismal prognosis [7]. In a French study from 2009 on adults, a corresponding analysis in contrast showed no difference in survival between patients with primary and secondary ES [19].

In contrast our results confirm that ITP in children has a very good prognosis, which correlates with previous results [20, 21].

## Treatment

Treatment of ES in children is generally challenging [3, 14]. The cornerstone in front-line treatment is steroids [14], especially in cases where AIHA is clinically dominant while IVIG can be preferred in cases with pronounced thrombocytopenia [3]. However, many patients need second-line treatments (e.g. rituximab and mycophenolate mofetil) [22], and there is a

general lack of randomized clinical trials investigating optimal treatment strategies. The heterogeneity of mechanisms underlying ES including secondary causes, requires the diagnostic work-up becomes even more thorough in order to individualize treatment [3, 6].

## Time trends

During our study period both ES incidence and prevalence increased although results are based on small numbers, and ES frequency measures are imprecise. The reasons for increasing frequencies remain elusive. They may, however, reflect advances in diagnostic technologies, as well as an increased awareness on ES and its severity from clinicians.

A general increase in incidence of autoimmune diseases have been observed over the last decades, which may also influence the ES frequency in children [23]. Explanations to this general increase probably includes both genetic and environmental contributing factors [24]. Environmental factors such as pesticides and solvents, and antibiotics interfering with microbiotic flora in the gut, are believed to contribute significantly to the increase in autoimmune diseases [24, 25]. Others suggest that air pollution causes oxidative stress and interference with the immune system, leading to activation and dysregulation of B-cells and the production of auto-antibodies [26]. These factors, combined with increasing urbanization and hence some environmental exposures in the Western industrialized world may elevate the autoimmune disease frequencies [13]."

Most splenectomies were performed before the year 2000. This is probably due to limited medical treatments options in this period (e.g. rituximab), and corresponds with splenectomy being a more frequent treatment early in our study period [18]. However, our data are based on few numbers of patients. Splenectomy is usually reserved as a 3$^{rd}$ or later line therapy in pediatric AIHA, ITP or ES, and with achieved complete remission rates and response durations greatly depending on disease with the poorest responses in ES [3, 18]

## Limitations of the study

We define secondary ES as the occurrence of associated diagnosis before or up to 1 year after diagnosis. However, secondary cases of ES may first be apparent up to several years after ES diagnosis [6]. In another study, diagnoses associated with ES were included from 6 years before up to 15 years after and was found to be mostly infections, immunodeficiency syndromes, and autoimmune diseases [6]. Possible underlying genetically conditioned immune disorders and differential diagnoses to ES (e.g. autoimmune lymphoproliferative syndrome (ALPS) and common variable immunodeficiency (CVID)), and the potential under diagnosis of these, especially in the beginning of our study period, could contribute to ES being misclassified as primary instead of secondary [6]. This could indicate that with longer follow-up, the proportion of ES patients with underlying associated diseases increases.

Sequence of AIHA and ITP diagnoses may also affect registration, since AIHA relies on haemolysis with a positive direct anti-globulin test, and ITP is a diagnosis of exclusion usually preceded by a longer diagnostic process.

## Conclusion

We conclude that ES in children below 13 years of age is a rare disorder with increasing incidence and prevalence over time, and associated with elevated mortality. Since ES is an orphan disease, ES in children should be reported to International prospective registers in order to inform and improve health care and to improve prognosis.

## Acknowledgments

The authors would like to thank Cathrine Fox Maule at Statistics Denmark for help with defining and hosting data.

## Author Contributions

**Conceptualization:** Nikolaj Mannering, Dennis Lund Hansen, Henrik Frederiksen.

**Data curation:** Nikolaj Mannering, Dennis Lund Hansen, Henrik Frederiksen.

**Formal analysis:** Nikolaj Mannering, Dennis Lund Hansen, Henrik Frederiksen.

**Funding acquisition:** Dennis Lund Hansen, Henrik Frederiksen.

**Methodology:** Nikolaj Mannering, Dennis Lund Hansen, Henrik Frederiksen.

**Project administration:** Nikolaj Mannering, Dennis Lund Hansen, Henrik Frederiksen.

**Software:** Dennis Lund Hansen.

**Supervision:** Dennis Lund Hansen, Henrik Frederiksen.

**Writing – original draft:** Nikolaj Mannering, Dennis Lund Hansen, Henrik Frederiksen.

**Writing – review & editing:** Nikolaj Mannering, Dennis Lund Hansen, Henrik Frederiksen.

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
