## [Decision Letter · Decision Letter 0]

27 Jan 2020

PONE-D-20-01354

Evans syndrome in children below 13 years of age – a nationwide population-based cohort study

PLOS ONE

Dear Dr Mannering,

Thank you for submitting your manuscript to PLOS ONE. After careful consideration, we feel that it has merit but does not fully meet PLOS ONE’s publication criteria as it currently stands. Therefore, we invite you to submit a revised version of the manuscript that addresses the points raised during the review process.

We would appreciate receiving your revised manuscript by Mar 12 2020 11:59PM. To enhance the reproducibility of your results, we recommend that if applicable you deposit your laboratory protocols in protocols.io, where a protocol can be assigned its own identifier (DOI) such that it can be cited independently in the future. For instructions see: http://journals.plos.org/plosone/s/submission-guidelines#loc-laboratory-protocols

We look forward to receiving your revised manuscript.

Kind regards,

David Meyre

Academic Editor

PLOS ONE

Journal Requirements:

2. In the ethics statement in the manuscript and in the online submission form, please provide additional information about the patient records used in your retrospective study. Specifically, please ensure that you have discussed whether all data were fully anonymized before you accessed them and/or whether the IRB or ethics committee waived the requirement for informed consent. If patients provided informed written consent to have data from their medical records used in research, please include this information.

4. Thank you for stating the following in the Competing Interests/Financial Disclosure section:

This project was part of a PhD fellowship (DLH), which was supported by the University of Southern Denmark (SDUSF-2015-202-(459)) https://www.sdu.dk/en/, The Region of Southern Denmark (16/13496) https://www.regionsyddanmark.dk/wm228983, and study grants from Alexion Pharma Nordic AB https://alexion.com/SelectCountry/Denmark, The A.P. Møller and Chastine Mc-Kinney Møller Foundation (17-L-0334) https://www.apmollerfonde.dk/ansoegning/laegefonden/, and Novartis Healthcare https://www.novartis.com/.

None of the authors have any other disclosures.

We note that you received funding from a commercial source: Alexion Pharma Nordic and Novartis Healthcare.

Reviewers' comments:

Reviewer's Responses to Questions

**Comments to the Author**

1. Is the manuscript technically sound, and do the data support the conclusions?

Reviewer #1: Yes

Reviewer #2: Yes

2. Has the statistical analysis been performed appropriately and rigorously? 

Reviewer #1: Yes

Reviewer #2: Yes

3. Have the authors made all data underlying the findings in their manuscript fully available?

Reviewer #1: Yes

Reviewer #2: Yes

4. Is the manuscript presented in an intelligible fashion and written in standard English?

Reviewer #1: Yes

Reviewer #2: Yes

5. Review Comments to the Author

Reviewer #1: The paper is very interesting and I compliment the authors for such a thorough investigation and long-term follow-up.

I have some suggestions for improvement, all of which cover minor issues.

Abstract, “The epidemiology and prognosis of Evans syndrome in patients above 13 years of age has recently been investigated”; this is not reported in the following text, and there is no clear reference; I guess that the authors refer to reference 6; anyway, the authors do not compare their present results to the previuos ones.

Introduction, “Here we assess both frequency and prognosis in Danish children below 13 years of age with ES”; why below 13 years? Please detail such a choice.

Methods, “ES was defined as the cumulative registration of ITP and AIHA in the same patient”: it should be emphasized that the diagnosis of ITP and AIHA does not necessarily be simultaneous.

Results, “in the period 1980-2016” In the Methods it was reported 1981-2015.

Results, “Splenectomy was performed in four (19%) patients” should be postponed; I ould rather report the epidemiological data first, and the clinical data thereafter.

Table, in the first column the “(%)” can be misleading, since what the other columns report, within parentheses, are the confidence intervals

Results, “four respectively one patient died”: I did not catch the meaning; furthermore, when reporting deaths, the authors report only percentages; it would help the comprehension to have absolute numbers

Discussion: the only therapy included in the whole paper is splenectomy, that has been considered an indicator of need for “stronger” treatments; however, the paper would have a deeper and sharper meaning if the severity of the disease and the challenge of the therapy would have been discussed.

Also, due to the rarity of the condition, some recent and insightful papers should be included in the reference list: Miano M et al. Mycophenolate mofetil for the treatment of children with immune thrombocytopenia and Evans syndrome. A retrospective data review from the Italian association of paediatric haematology/oncology. Br J Haematol. 2016 Nov;175(3):490-495

Miano M. How I manage Evans Syndrome and AIHA cases in children. Br J Haematol. 2016 Feb;172(4):524-34.

Ladogana S et al.Diagnosis and management of newly diagnosed childhood autoimmune haemolytic anaemia. Recommendations from the Red Cell Study Group of the Paediatric Haemato-Oncology Italian Association. Blood Transfus 2017; 15(3):259-67

Reviewer #2: The manuscript PONE-D-20-01354 titled ‘Evans syndrome in children below13 years of age-a nationwide population-based cohort study” describes the incidence, prevalence and hazard ratio for death of children with Evans syndrome in Denmark. Since this is one of the very few registry-based studies on this very rare autoimmune disorder, it deserves consideration of publication. Of course, most, if not all of the described results have previously been shown by others. For example, it is well-known that Evans syndrome has worse prognosis and increased mortality compared to AIHA or ITP alone. Although the authors should be congratulated for the study, Denmark is a very small country with just 5,6 million people. A Scandinavia-based study that would include patients from Denmark, Sweden, Norway and Finland (estimated population of 26 million people), would add strength to the study’s conclusions, since Evans syndrome is so rare in childhood that e.g., only 2 deaths were recorded over the entire study period. Moreover, several points of the manuscript need clarifications and revisions.

First, the definition of secondary Evans syndrome that was used is extremely problematic. One year of follow-up after Evans’ syndrome diagnosis is completely inadequate to exclude with confidence cases of secondary disease. Although the authors refer to this limitation in the discussion, this is done at the wrong place (lines 125-127). A dedicated paragraph with the limitations of the study should be added just prior to the concluding paragraph of the discussion.

Second, the authors fail to discuss one of the main findings of the study, i.e., the fact that Evans syndrome in Denmark is more common in boys (2 to 1 ratio) like it is in France (reference No. 6). The authors should elaborate on this in the discussion, since indeed Evans syndrome appears to be more common in males during childhood but more common in females during adulthood (as it is the case with other autoimmune diseases). In addition, potential explanations for this sex difference by age should be offered.

Third, the authors should elaborate on the potential reasons for the increased incidence and prevalence of Evans syndrome with time, i.e., they should give some plausible explanations for the almost 2.5 times higher incidence and prevalence of Evans syndrome in 2015 compared to 1990.

Fourth, it would be interesting to study the year in which the 4 splenectomies for patients with Evans syndrome were performed. In the 1990s splenectomy was likely more commonly employed for Evans syndrome , AIHA or ITP compared to 2015, but the relevant data should be provided.

Fifth, more references are required, and a more in-depth discussion is required. Finally, there are few typos and missing words throughout the manuscript. For example, in the abstract (line 20), the word register should be replaced with the word registry, since the described study is a registry-based not a register-based study. Also, in line 64 the word between is missing (All patients with ES below 13 years of age diagnosed between 1981-2015 were included in analyses). Also, in lines 138-139, the word is should be added: Despite of this, we managed to identify only 21 patients emphasizing that pediatric ES is a very rare entity.

6. PLOS authors have the option to publish the peer review history of their article (what does this mean?). If published, this will include your full peer review and any attached files.

Reviewer #1: Yes: Giovanna Russo

Reviewer #2: Yes: Elpis Mantadakis, MD, PhD

---

## [Author Response · Author response to Decision Letter 0]

10 Mar 2020

Dear Editor & Reviewers,

Thank you for your constructive revision of our submitted manuscript.

We are happy to return a revised edition, including a Rebutal Letter with point-by-point answers to your raised concerns regarding our manuscript. Please refer to this for details.

We are looking forward to your response.

best regards

Nikolaj Mannering, M.D.

Odense University Hospital

Denmark

---

## [Editor Report · Decision Letter 1]

20 Mar 2020

Evans syndrome in children below 13 years of age – a nationwide population-based cohort study

PONE-D-20-01354R1

Dear Dr. Mannering,

We are pleased to inform you that your manuscript has been judged scientifically suitable for publication and will be formally accepted for publication once it complies with all outstanding technical requirements.

With kind regards,

David Meyre

Academic Editor

PLOS ONE
---

## [Editor Report · Acceptance letter]

24 Mar 2020

PONE-D-20-01354R1 

Evans syndrome in children below 13 years of age – a nationwide population-based cohort study 

Dear Dr. Mannering:

I am pleased to inform you that your manuscript has been deemed suitable for publication in PLOS ONE. Congratulations! Your manuscript is now with our production department. 

With kind regards,

on behalf of

Dr David Meyre 

Academic Editor

PLOS ONE